

# OsteoporosAtlas: a human osteoporosis-related gene database

Xun Wang[1,*], Lihong Diao[1,*], Dezhi Sun[1], Dan Wang[1], Jiarun Zhu[1,4], Yangzhige He[1,3], Yuan Liu[1], Hao Xu[1], Yi Zhang[1,4], Jinying Liu[2], Yan Wang[1], Fuchu He[1], Yang Li[1] and Dong Li[1]

[1] State Key Laboratory of Proteomics, Beijing Proteome Research Center, National Center for Protein Sciences, Beijing Institute of Lifeomics, Beijing, China
[2] School of Traditional Chinese Medicine, Beijing University of Chinese Medicine, Beijing, China
[3] Central Research Laboratory, Peking Union Medical College Hospital, Chinese Academy of Medical Sciences & Peking Union Medical College, Beijing, China
[4] College of life Sciences, Hebei University, Baoding, China
[*] These authors contributed equally to this work.

## ABSTRACT

**Background**. Osteoporosis is a common, complex disease of bone with a strong heritable component, characterized by low bone mineral density, microarchitectural deterioration of bone tissue and an increased risk of fracture. Due to limited drug selection for osteoporosis and increasing morbidity, mortality of osteoporotic fractures, osteoporosis has become a major health burden in aging societies. Current researches for identifying specific loci or genes involved in osteoporosis contribute to a greater understanding of the pathogenesis of osteoporosis and the development of better diagnosis, prevention and treatment strategies. However, little is known about how most causal genes work and interact to influence osteoporosis. Therefore, it is greatly significant to collect and analyze the studies involved in osteoporosis-related genes. Unfortunately, the information about all these osteoporosis-related genes is scattered in a large amount of extensive literature. Currently, there is no specialized database for easily accessing relevant information about osteoporosis-related genes and miRNAs.
**Methods**. We extracted data from literature abstracts in PubMed by text-mining and manual curation. Moreover, a local MySQL database containing all the data was developed with PHP on a Windows server.
**Results**. OsteoporosAtlas (http://biokb.ncpsb.org/osteoporosis/), the first specialized database for easily accessing relevant information such as osteoporosis-related genes and miRNAs, was constructed and served for researchers. OsteoporosAtlas enables users to retrieve, browse and download osteoporosis-related genes and miRNAs. Gene ontology and pathway analyses were integrated into OsteoporosAtlas. It currently includes 617 human encoding genes, 131 human non-coding miRNAs, and 128 functional roles. We think that OsteoporosAtlas will be an important bioinformatics resource to facilitate a better understanding of the pathogenesis of osteoporosis and developing better diagnosis, prevention and treatment strategies.

Corresponding authors
Yang Li, liyang_bprc@163.com
Dong Li, lidong.bprc@foxmail.com

## INTRODUCTION

Osteoporosis is a common skeletal disease under strong genetic control (*Ralston & De Crombrugghe, 2006*), whose characteristics are low bone mineral density (BMD), microarchitectural deterioration of bone tissue and increased risk of fracture (*Kanis et al., 1994*). Osteoporosis is generally induced by various causes, such as age, sex, genetic factors, a range of hormones and environmental factors. Moreover, genetic factors not only have strong control over osteoporosis and its associated phenotypes, but may also influence sensitivity to hormonal and environmental factors (*Eisman, 1999*). Osteoporosis can be divided into primary, secondary, and idiopathic types. Primary osteoporosis is more common and mostly caused by old age and postmenopause (*Ji & Yu, 2015*). Secondary osteoporosis is most commonly caused or exacerbated by medication exposures or other disorders (*Stein & Shane, 2003*). Idiopathic osteoporosis is a type of rare osteoporosis that often occurs in young, healthy individuals who are not postmenopausal or have other, identifiable secondary causes of osteoporosis (*Heshmati & Khosla, 1998*). The clinical outcomes of osteoporosis are fracture of hip, wrist, spine and other types. Hip fracture, an international barometer of osteoporosis, is the most severe osteoporotic fracture, 10–20% more women with hip fracture die than expected for age within the first year, and the mortality rate in men with hip fracture is even higher (*Cummings & Melton, 2002*). Menopausal estrogen deficiency is one of the most important reasons why women are more susceptible to osteoporosis. In addition, men generally have greater cortical mass and larger bone size than women (*Richelson et al., 1984*). Alarmingly, osteoporosis affects over 200 million individuals worldwide and the incidence of osteoporotic fractures is expected to rise by 50% over the next decade (*Burge et al., 2007*). With the aging of the population in recent years, osteoporosis is becoming an increasingly huge public health issue associated with increased mortality and morbidity.

Osteoporosis is a polygenic disorder, generally determined by the combined effects of several genes and environmental factors. Osteoporosis rarely occurs as the result of single gene mutations. Twin and family studies have shown that genetic factors are extremely important in the regulation of bone density, skeletal geometry, ultrasound properties of bone, bone turnover, and risk of osteoporosis (*Stewart & Ralston, 2000*). Identification of specific loci or genes determining osteoporosis will contribute to a better understanding of the pathogenesis of osteoporosis and developing better diagnosis, prevention and treatment strategies. The current efforts to identify osteoporosis loci or genes have mainly focused on three approaches: animal models, candidate gene approach and genome-wide scans (*Huang, Recker & Deng, 2003*). Genome-wide association studies have identified over 300 loci associated with BMD. However, little is known of how most causal genes interact with each other and the mechanisms to cause osteoporosis (*Al-Barghouthi & Farber, 2018*). Recently, some studies have shown that how causal genes contribute to the pathogenesis of osteoporosis. For example, the research has shown that Siglec-15 gene-deficient mice exhibit mild osteoporosis and Siglec-15 gene is involved in osteoclast differentiation induced by estrogen deficiency, which suggests Siglec-15 gene is a promising drug therapy target for postmenopausal osteoporosis and age-related osteoporosis (*Kameda et al.,*

*2015*). C-Abl gene and Atm (ATM Serine/Threonine Kinase) gene result in osteoporosis by positively regulating osteoblast differentiation and bone formation, moreover mice lacking either of them shows osteoporosis, while P53 gene inhibits osteoporosis, which negatively regulates osteoblast differentiation and bone formation, and the knockout mouse shows osteosclerosis (*Wang & Li, 2007*). Sclerostin (SOST gene encoded) is an antagonist of WNT/β-catenin signaling (canonical WNT pathway), which predominantly regulates osteoblast differentiation and plays important roles in regulating bone formation. Inactivating monoclonal antibodies against SOST, a inhibitor of the negative regulation of WNT/β-catenin signaling, has been shown to be a candidate for the prevention and treatment of osteoporosis (*Rossini, Gatti & Adami, 2013*).

MicroRNAs (miRNAs), a class of small (∼22 bp) nucleotides, are single-stranded noncoding RNAs that regulate the expression of target genes by binding to their 3′-untranslated region (*Carthew & Sontheimer, 2009*). miRNAs play critical roles in the regulation of various biological processes by targeting mRNAs, including cellular differentiation and proliferation, apoptosis, and tissue development (*Kim, Han & Siomi, 2009*). An imbalance between osteoblastic bone formation and osteoclastic bone resorption plays a fundamental role in osteoporosis pathogenesis (*Teitelbaum, 2000*). Recent studies discovered that miRNAs play an important role in the subtle equilibrium between bone formation and bone resorption by targeting various genes to regulate osteoblast and osteoclast differentiation and function (*Ji, Chen & Yu, 2016*). For example, *Zhao et al. (2015)* have found that hsa-mir-21 overexpression aggravates osteoporosis by targeting RECK. Hsa-mir-34a is a novel and pivotal suppressor of osteoclastogenesis and bone resorption, which blocks osteoporosis by inhibiting osteoclastogenesis and Tgif2 (*Krzeszinski et al., 2014*). Hsa-mir-133a was shown to directly target Runx2 gene 3′-UTR when overexpressed in MC3T3, an osteoblast cell line. In addition, hsa-mir-133a was also shown to negatively regulate three potential osteoclast-related target genes, CXCL11, SLC39A1 and CXCR3. MiRNA expression analysis in human circulating monocytes shows that hsa-mir-133a is a promising biomarker for postmenopausal osteoporosis (*Wang et al., 2012*).

Currently, therapeutic drugs for osteoporosis include hormone replacement therapy, calcitonin, selective estrogen receptor, and bisphosphonates, etc. However, due to side-effects and high price, drug selection for osteoporosis is still limited (*Martin & Sims, 2005*). Therefore, research for identifying osteoporosis-related genes and their pathogenic mechanisms is greatly significant, which promote a better understanding of the pathogenesis of osteoporosis, identification of novel biomarkers, discovery of therapeutic targets and accurate treatment strategies. However, abundant information of previously confirmed osteoporosis-related genes is scattered in a large amount of extensive literature. Moreover, there is no specialized knowledge base dedicated to osteoporosis-related genes collection, which makes it difficult to understand the pathogenesis of osteoporosis and develop new drug targets for osteoporosis. In order to address this obstacle, we established OsteoporosAtlas 1.0 (http://biokb.ncpsb.org/osteoporosis/) by literature-mining and manual curation, which currently contains 617 human osteoporosis-related genes, 131 human osteoporosis-regulated miRNAs, 84 biological process analysis, and 128 functional

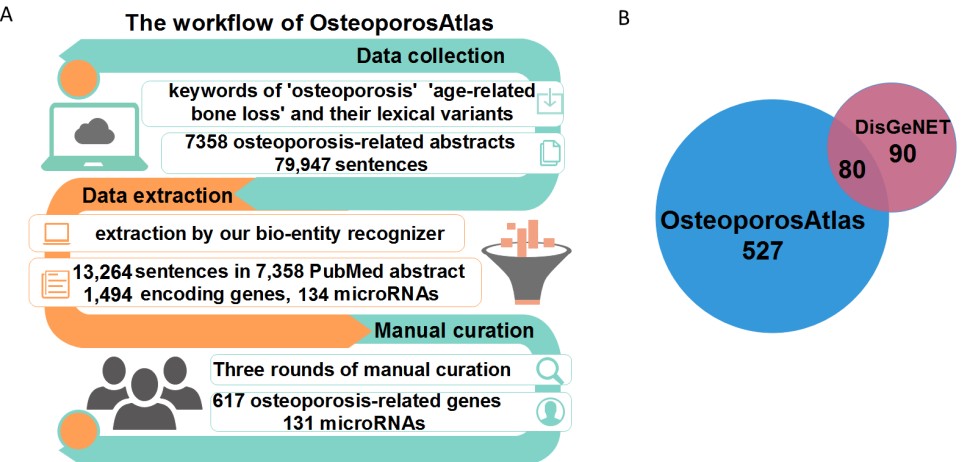

**Figure 1** **The construction workflow of OsteoporosAtlas 1.0 database.** (A) The workflow of Osteo-porosAtlas 1.0 database construction. (B) The comparison of osteoporosis-related genes between Osteo-porosAtlas 1.0 database and DisGeNET database.

roles. Users can retrieve, browse and download osteoporosis-related genes and relevant information from OsteoporosAtlas. Moreover, gene ontology analysis, pathway analysis, and SNP term analysis were integrated into OsteoporosAtlas. In summary, we believe that OsteoporosAtlas will provide a comprehensive osteoporosis research resource for the pathogenesis of individual cases, discovery of drug target, novel diagnostic biomarker, individual treatment methods and precision medication.

# MATERIALS & METHODS

## Text mining and manual curation

We developed an ontology-based bio-entity recognizer (see the Appendix for details). Abstract data of CRAFT 2.0 (*Verspoor et al., 2012*) corpus was used as the independent test set to evaluate the performance of our bio-entity recognizer by identifying gene/protein based on Protein Ontology (*Natale et al., 2010*). After evaluation, the precision (the number of correct entities identified divided by the number of entities identified), recall (the number of correct entities identified divided by the number of entities in the sample), F-measure (the harmonic mean of precision and recall) of our bio-entity recognizer for identifying gene/protein were respectively 0.959, 0.802 and 0.874, indicating it is of great performance for entity recognition (both the evaluation program and result was available at http://biokb.ncpsb.org/osteoporosis/Public/file/craft_test.rar). Therefore, we performed a comprehensive search for osteoporosis-related literature abstracts in PubMed to extract osteoporosis-related genes by using the bio-entity recognizer. Specifically, it is divided into three steps (Fig. 1A):

First, data collection: we collected 79,947 sentences in 7,358 abstracts containing the keywords of "osteoporosis", "osteoporoses", "bone loss, age-related", "age-related bone loss", and "perimenopausal bone loss".

Second, candidate data extraction: 1,628 osteoporosis-related candidate genes/microRNAs and 13,264 candidate evidence sentences (Table S1) were extracted by using our bio-entity recognizer.

Third, manual curation: our experts manually curated all candidate genes and evidence sentences, 527 encoding genes and 131 miRNAs were finally confirmed as human osteoporosis-related genes. In addition, we integrated osteoporosis-related genes confirmed in DisGeNET (*Piñero et al., 2015*) (Fig. 1B), an integrated comprehensive platform with information about human disease-associated genes and variants. In summary, our database consists of 617 osteoporosis-related encoding genes, 131 osteoporosis-regulated miRNAs (non-coding RNA).

## Gene annotation

In order to facilitate the deep interpretation of the association between osteoporosis and related genes, we annotated each gene by the basic gene annotation files (''gene2refseq'') from NCBI FTP site with the information of gene synonyms symbol, genetic location, gene full name, chromosome, gene type, reference sequence information and chromosomal location. We obtained gene ontology annotation for each gene from the Gene Ontology Annotation database (GOA) (*Camon et al., 2004*) and gene-pathway mapping relationship from the Reactome database (*Fabregat et al., 2015*). The database of short genetic variation (dbSNP) (*Sherry et al., 2001*) was used to map SNPs to gene by the publications' PMIDs (PubMed sole Identifier), which makes it easier to access to genome-wide association study and helps researchers to improve the understanding of the genetic background of osteoporosis. Mapping and annotating are done by using public databases of Entrez gene (*Maglott et al., 2005*), Ensembl (*Flicek et al., 2010*), UniProt (*Bairoch et al., 2005*), Antibodypedia (*Björling & Uhlén, 2008*) and neXtProt (*Lane et al., 2011*).

## RESULTS

### Database service and implement

All osteoporosis-related genes and their relevant information were deposited into a local MySQL database. The website of OsteoporosAtlas (http://biokb.ncpsb.org/osteoporosis/) was developed with PHP on a Windows server. All data about OsteoporosAtlas is accessible for all users without registration or login.

### Database navigation and search

In order to provide a user-friendly web interface for searching and browsing, we designed five sections of functions for our database (Fig. 2A) and provide three query approaches.

For the search by gene name, users can input a gene symbol in the ''Gene Name'' search box (Fig. 2B). After clicking the ''search'' button, user can get detailed gene annotation for osteoporosis (Fig. 2D), including functional role (whether this gene is a biomarker/drug target), supporting literature evidence, SNP information from the database of short genetic variation (dbSNP), gene ontology (GO) terms from Gene Ontology Annotation database (GOA), the protein description from database UniProtKB, pathway terms from Reactome analysis, gene expression information from Expression

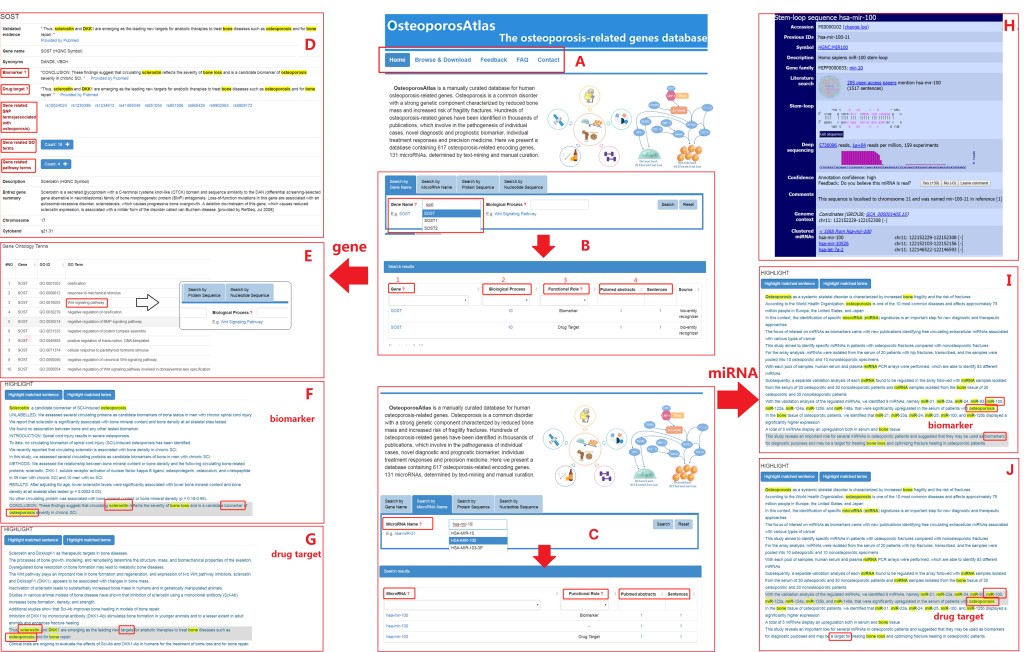

**Figure 2   The outline for searching in OsteoporosAtlas 1.0 website.** (A) Five functional sections of the database. (B) Users can submit a gene name to the "Gene Name" search box and the search result will be returned, including the information of Gene, related biological processes, functional role, and evidence. (D) After clicking the gene name, users can get more specific information about this gene on the detailed page. (E) After clicking on the number of biological processes, users can scan the biological processes involved by this gene. (F, G) After clicking the number of evidence, the original abstract will be displayed with highlighted matched sentence and matched keywords. (C) Users can submit a microRNA name to the "MicroRNA Name" search box and the search result will be returned, including the information of microRNA, functional role and support evidence. (H) After clicking the microRNA name, users can get more specific information about this microRNA in the detailed page. (I, G) After clicking the number of the evidence, the original abstract will be displayed with highlighted matched sentence and matched keywords.

Atlas, protein expression information from Human Protein Atlas (HPA), and the regulated miRNAs information from miRBase database (*Kozomara & Griffiths-Jones, 2013*). Clicking the number of abstracts as evidence will lead to the detailed info of the evidence abstracts (Figs. 2F and 2G), and clicking the number of the involved biological process will lead to the view of the involved biological processes (Fig. 2E).

The section of 'Browse & Download' provides four different approaches, including genes, miRNAs, biological processes and functional roles. The current version of database contains 617 genes, 131 miRNAs, 84 biological processes and 128 functional roles. All the information can be downloaded. In the section of "Feedback", we provide a submitting function. If users found osteoporosis-related genes which are absent from our database, they can submit this gene to our database.

## Database use case

Users can find *SOST* gene by searching gene name, protein sequence or nucleotide sequence in our database. Four functional sections are provided in the search results interface
(Fig. 2B): Relevant information about gene interface (Fig. 2D) provides functional role (whether this gene is a biomarker/drug target), supporting literature evidence, relevant information about genome-wide association study from the database of short genetic variation (dbSNP), gene ontology (GO) terms from Gene Ontology Annotation database, the protein description from UniProtKB database, pathway terms from Reactome analysis, gene expression information from Expression Atlas, protein expression information from Human Protein Atlas (HPA). Evidence interface (Figs. 2F and 2G) provides the literature which supports the association of *SOST* gene with osteoporosis. Biological process interface (Fig. 2E) provides gene ontology terms associated with *SOST* gene. Searching the name of the biological process can browse the genes in this biological process.

Users can find hsa-mir-100 by searching miRNA name in our database. Three functional sections are provided in the search results interface (Fig. 2C). Clicking hsa-mir-100 will jump to miRBase database, an online resource for miRNA sequence and annotation (Fig. 2H). Evidence interface provides the literature which supports the association of hsa-mir-100 with osteoporosis (Figs. 2I and 2J).

## DISCUSSION

The systematic collection of the OsteoporosAtlas 1.0 database provides an overview of human osteoporosis-related genes. Reactome analysis shows that these genes are involved in pathways of signal transduction, immune system, gene expression (transcription), extracellular matrix organization, etc (Fig. 3A). Gene ontology analysis using Panther indicates that the most common class of osteoporosis-related genes belongs to signaling molecule, followed by hydrolase/nucleic acid binding, receptor/transcription factors, enzyme modulator, oxidoreductase, transferase, transporter, etc (Fig. 3B). The Reactome and gene ontology analysis could help biologists to better understand the functional relevance of these genes and guide the experiment. In addition, the existence of many potential pharmacological targets in these pathways we analyzed makes it attractive for osteoporosis drugs discovery. All the results show the value of our collection, demonstrating that the OsteoporosAtlas 1.0 database will greatly benefit exploration of pathogenesis and treatment of osteoporosis.

OsteoporosAtlas 1.0 also has community curation feature. All users can log in to provide their feedback, confirm or reject the evidence sentence by clicking the "Yes" or "No" button. We will update the database periodically according to users' feedbacks.

We identified 94 drug target genes with definite literature evidences in our database. The existing osteoporosis treatment drugs mainly delay bone loss by inhibiting bone resorption or promoting bone formation. However, due to the high price and side effects, the osteoporosis drugs promoting bone formation without side-effects are still in the basic clinical research stages. So our database will help discover new drugs for the treatment of osteoporosis.

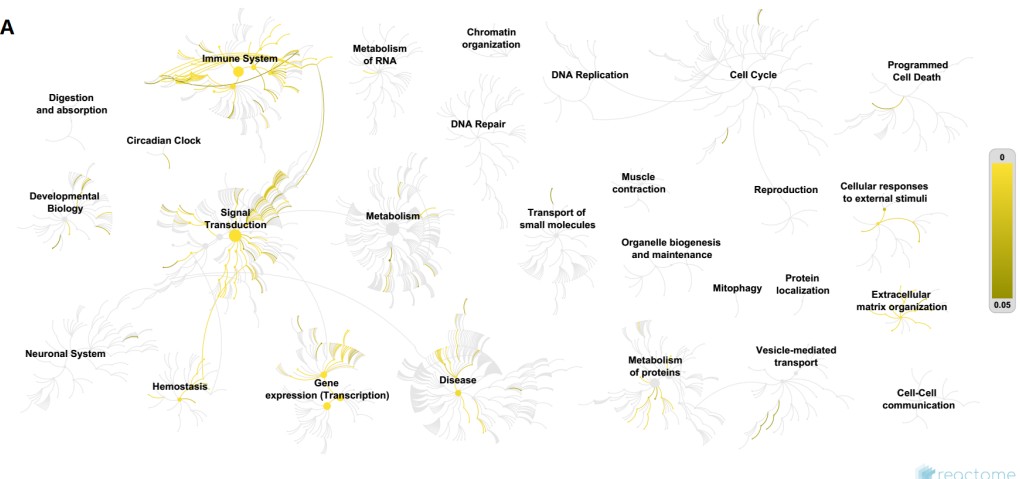

**B**     **Protein classification of 617 osteoporosis-related genes**

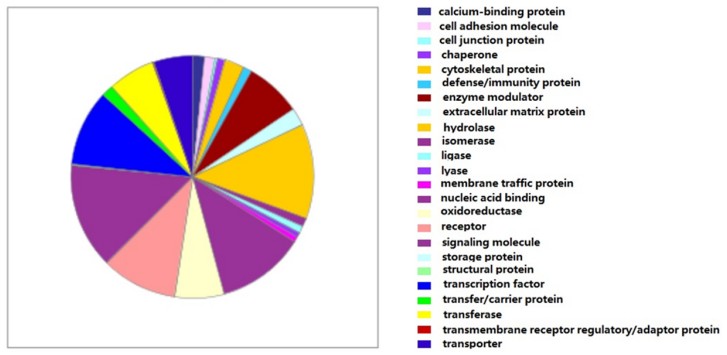

- calcium-binding protein
- cell adhesion molecule
- cell junction protein
- chaperone
- cytoskeletal protein
- defense/immunity protein
- enzyme modulator
- extracellular matrix protein
- hydrolase
- isomerase
- ligase
- lyase
- membrane traffic protein
- nucleic acid binding
- oxidoreductase
- receptor
- signaling molecule
- storage protein
- structural protein
- transcription factor
- transfer/carrier protein
- transferase
- transmembrane receptor regulatory/adaptor protein
- transporter

**Figure 3**   **Bioinformatics analysis of genes associated with osteoporosis.** (A) Biological pathway analysis using Reactome (http://www.reactome.org/). (B) Protein class analysis using PANTHER (http://www.pantherdb.org).

## CONCLUSIONS

In conclusion, OsteoporosAtlas 1.0 collects 617 osteoporosis-related encoding genes and 131 miRNAs, which is the first database specially to present a comprehensive list of osteoporosis-related genes obtained from published literature. We believe that OsteoporosAtlas 1.0 will be widely used as it can provide facilities for scientists and clinicians in searching the literature on osteoporosis-related genes and their function in diseases.

## ACKNOWLEDGEMENTS

We would like to thank the anonymous reviewers for their useful comments and suggestions on this manuscript.

# APPENDIX

The detailed description of methods:

(1) Construction and processing of vocabulary

It is necessary to construct different types of vocabularies for natural language processing and semantic recognition, including stop words, high frequency words, domain words and abbreviations. Stop words are mainly from PubMed, MetaMap (*Aronson, 2001*), and Python NLTK (*Loper & Bird, 2002*). High frequency words are mainly from PIR (Protein Information Resource (*Barker et al., 2000*)) Common English, General Names, Common English of PIR iProLINK (*Hu et al., 2004*) Entity Recognition, and Google 1000 English. Domain words are mainly from the gene/protein, compound, enzyme, and other biochemical naming dictionary in PIR. Abbreviations are mainly from the open source software BADREX (*Gooch, 2012*) in Github.

(2) Collection and processing of literature

The PubMed abstracts were downloaded in XML format by NCBI E-Utilities API (*Sayers, 2010*) and parsed by Python script. The database currently contains approximately 26 million abstracts.

(3) Collection and processing of ontology

A high-quality dictionary of human genes/proteins and their synonyms is mainly based on Protein Ontology (*Natale et al., 2010*) and bioThesaurus. And the human protein ID of UniProtKB is used as a standardized ID.

(4) Entity recognition

Indexer (Python script) is used to build indexes for the PubMed abstracts. Analyzer (Python script) is used to perform synonym expansion on the ontology. Indexes and synonyms were combined to perform span near match and entity recognition.

## Funding

This work is funded by the National Natural Science Foundation of China (31871341, 31601064), the State Key Laboratory of Proteomics (SKLP-K201702), the Innovation Project (16CXZ027), the Beijing Nova Program (Z171100001117117) and the Program of Precision Medicine (2016YFC0901905). The funders had no role in study design, data collection and analysis, decision to publish, or preparation of the manuscript.

## Grant Disclosures

The following grant information was disclosed by the authors:
National Natural Science Foundation of China: 31871341, 31601064.
State Key Laboratory of Proteomics: SKLP-K201702.
Innovation Project: 16CXZ027.
Beijing Nova Program: Z171100001117117.
Program of Precision Medicine: 2016YFC0901905.

![PeerJ]

## Competing Interests

The authors declare there are no competing interests.

## Author Contributions

- Xun Wang performed the experiments, analyzed the data, contributed reagents/materials/analysis tools, prepared figures and/or tables, authored or reviewed drafts of the paper, approved the final draft.
- Lihong Diao and Yuan Liu performed the experiments, contributed reagents/materials/analysis tools, approved the final draft.
- Dezhi Sun performed the experiments, analyzed the data, contributed reagents/materials/analysis tools, approved the final draft.
- Dan Wang and Yangzhige He performed the experiments, analyzed the data, approved the final draft.
- Jiarun Zhu, Hao Xu and Yi Zhang performed the experiments, authored or reviewed drafts of the paper, approved the final draft.
- Jinying Liu and Yan Wang performed the experiments, approved the final draft.
- Fuchu He conceived and designed the experiments, approved the final draft.
- Yang Li conceived and designed the experiments, authored or reviewed drafts of the paper, approved the final draft.
- Dong Li conceived and designed the experiments, analyzed the data, authored or reviewed drafts of the paper, approved the final draft.

## Data Availability

The raw data are provided in Table S1. The database can be found at http://biokb.ncpsb.org/osteoporosis/.

## Supplemental Information

Supplemental information for this article can be found online at http://dx.doi.org/10.7717/peerj.6778#supplemental-information.

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
