# Peer review of "OsteoporosAtlas: a human osteoporosis-related gene database"

_PeerJ, doi:10.7717/peerj.6778_

## Round 0.1 · original submission · Major Revisions

Please address all the critical issues raised by all 4 reviewers and amend your manuscript accordingly

Reviewer 1 ·

Basic reporting

OsteoporosAtlas: a human osteoporosis-related genes database by Wang et al. is a comprehensive database of 617 osteoporosis-related genes and 131 osteoporosis regulated miRNAs. It provides a research resource for the pathogenesis of the disease for individual cases, the discovery of drug target, biomarkers and individual treatment methods and medication. I commend the authors for building such an extensive database. In addition, the manuscript is clearly written in professional, unambiguous language. Having said that, I feel the authors did not cover the existing literature well enough and this should be improved before acceptance. I have these suggestions that may help to improve this issue.
1. Here the description and characteristics of osteoporosis have been oversimplified. It is a complex polygenic disease and genetic as well as environmental factors contribute to the cause of this disease. For more details, authors may follow these articles Huang QY et al. Osteoporos Int. 2003 and Cummings SR et al. Lancet. 2002.
2. Authors in line 38-39 categorized the cause of the disease into three types, however, they only described the primary type in the manuscript. Authors should also provide a brief description of the secondary and idiopathic types as well.
3. Line 47 is not clear to me, please elaborated with some evidence.
4. Authors should mention in detail how the present study will help to improve the understanding of the genetic background of this disease and how it fills the existing knowledge gap. I think including the genome-wide association study in understanding the genetics of osteoporosis will help to make a bridge between the existing knowledge and what this study offers to understand the disease.
5. Please provide a reference for the sentence in the line 166-167.

Experimental design

Relevance: Of sufficient interest
Originality: Moderately original
Clarity: Clear enough

Validity of the findings

Technical Correctness: Probably correct
Experimental Validation: Sufficient validation

Reviewer 2 ·

Basic reporting

N/A

Experimental design

N/A

Validity of the findings

N/A

Additional comments

The manuscript describes a database resource for osteoporosis related genes and microRNAs, collected using a text mining approach. The database has a very specific application and may be suitable for specific readership concerned about the particular disease. The database functionalities are not working now, please check all the links. Also, in the manuscript Results section, please add few example case studies from the database on both genes and microRNAs related to osteoporosis.

Reviewer 3 ·

Basic reporting

The manuscript is very well written. Some minor corrections are suggested as follows:
1. Line 42: “issue” instead of “issues”; “brings” instead of “bring”
2. Line 46, 63,93: Avoid sentence start with “And”
3. Line 48: “Lots of studies have shown that many encoding genes’ deficiency or
overexpression in mice cause the inhibition or promotion of osteoporosis”- confusing, consider re-write
4. Line 63: “plays” instead of “play”
5. Line 69: “increasing” instead of “increasingly”
6. Line 71: “aggravates” instead of “aggravate”
7. Line 77: “associated” not “associate”
8. Lines 82: “…which makes it difficult…” instead of “which is the difficulty”
9. Line 84: “contains” not “contain”
10. Line 113: What is the GOA database? Other abbreviations are defined
11. Line 158: “OsteoporosAtlas 1.0 also of the function of community curation”-meaning not clear , re-write
12. Line 163-164: “….the osteoporosis drugs promoting…” instead of “…the osteoporosis drugs by promoting bone…”
13. Also check spacing between words and brackets.

Comments about figures and supplemental files:
1. Figure 1A, 2, 3A: Font is very small.
2. No description of supplemental files in the text. Readers will not be aware of the existence of these files.
3. peerj-32180-Osteoporosis_gene.txt – check column alignment

Experimental design

No comments

Validity of the findings

No comments

Reviewer 4 ·

Basic reporting

The manuscript entitled “OsteoporosAtlas: a human osteoporosis-related gene database” reports the first osteoporosis database that catalogs the genes known to be associated with osteoporosis, a common bone disease. This database provides easy access to the information on osteoporosis-related genes to the scientific researchers and clinicians.
The article structure is appropriate according to the journal. However, authors fail to introduce the research problem to the broad readers for a clear understanding of their work.
Also, the manuscript is poorly-written and I strongly recommend the authors to take help from professional English language editors.

Experimental design

In the study, the authors used appropriate experimental strategy for the extraction of the data from the biomedical literature, however, the authors did not provide a complete detail description of the methods in the manuscript which needs to be addressed for better understanding to broad interest readers.

Validity of the findings

No comment

Additional comments

1. Please re-write the abstract section especially background and results section. In the background section, the research problem is not presented clearly. In the results section, highlight the fact that it is a first specialized database for the easy access of relevant information such as osteoporosis-related genes and miRNAs.
2. I suggest to re-write the introduction section of the manuscript. As it stands now, it reads poor and is difficult to follow. It should be clear and more general; discuss different causes of osteoporosis, its pathophysiology and role of miRNAs rather than including examples of different genes and miRNAs.
3. In materials and methods section, the authors did not introduce the terms such as whose precision, F-measure and recall against the CRAFT corpus for protein ontology-based gene/protein conformation for the readers to understand them clearly. Also, please explain the methods in detail.
4. Throughout the manuscript, I observed lot of grammatical errors, wrong usage of punctuations which makes it difficult for reader to interpret the actual meaning of the sentence.
5. In the results section, line 166-169 seems to be irrelevant. Also, the authors did not cite any references.
6. Specific remarks:
line 37-38: Re-write. Same as line 20-21 in the background section;
line 43: Include 2-3 sentences to mention the reasons why women are more prone to osteoporosis;
line 48-49: Rephrase the sentence, not clear;
line 77: change “associate” to “associated”;
line 81-82: Rephrase the sentence, difficult to interpret;
line 87: write “precision” instead of “precise”;
line 102-103: Rephrase;
line 106: change “finally” into “In summary, our database consists of”;
line 121: Remove “comma” from the text “their, functional”;
line 122-123: merge the two sentences;
line129: Rephrase the text from “after clicking the button of search,” to “after clicking the “search” button,”

---

## Round 0.2 · accepted · Accept

Both reviewers were satisfied with your revision.

# Reviewer 1 ·

Basic reporting

The manuscript is written in clear and unambiguous manner.

Experimental design

Original primary research within Aims and Scope of the journal.

Validity of the findings

Conclusions are well stated, linked to original research question & limited to supporting results.

Additional comments

I find the revised manuscript is suitable for publication in PEER J.

Reviewer 2 ·

Basic reporting

The authors have addressed all of my concerns. I am happy with their answers. I strongly recommend this article to publish.

Experimental design

N/A

Validity of the findings

N/A

Additional comments

The authors have addressed all of my concerns. I am happy with their answers. I strongly recommend this article to publish.